# A Mousepad Triboelectric-Piezoelectric Hybrid Nanogenerator (TPHNG) for Self-Powered Computer User Behavior Monitoring Sensors and Biomechanical Energy Harvesting

**DOI:** 10.3390/polym15112462

**Published:** 2023-05-26

**Authors:** Gang Jian, Ning Yang, Shangtao Zhu, Qingzhen Meng, Chun Ouyang

**Affiliations:** 1Shenzhen Institute of Advanced Electronic Materials, Shenzhen Institutes of Advanced Technology, Chinese Academy of Sciences, Shenzhen 518055, China; 2School of Materials Science and Engineering, Jiangsu University of Science and Technology, Zhenjiang 212100, China; zhushangtao1999@163.com (S.Z.); qingzhen_m@126.com (Q.M.); oyc1014@163.com (C.O.); 3Wuxi Hansu Technology Co., Ltd., 216 Xitai Road, Wuxi 214111, China

**Keywords:** triboelectric-piezoelectric hybrid nanogenerator (TPHNG), mousepad, film, surface charging, behavior monitoring, energy

## Abstract

Hybrid nanogenerators based on the principle of surface charging of functional films are significant in self-powering sensing and energy conversion devices due to their multiple functions and high conversion efficiency, although applications remain limited due to a lack of suitable materials and structures. Here, we investigate a triboelectric-piezoelectric hybrid nanogenerator (TPHNG) in the form of a mousepad for computer user behavior monitoring and energy harvesting. Triboelectric and piezoelectric nanogenerators with different functional films and structures work independently to detect sliding and pressing movements, and the profitable coupling between the two nanogenerators leads to enhanced device outputs/sensitivity. Different mouse operations such as clicking, scrolling, taking-up/putting-down, sliding, moving rate, and pathing can be detected by the device via distinguishable patterns of voltage ranging from 0.6 to 36 V. Based on operation recognition, human behavior monitoring is realized, with monitoring of tasks such as browsing a document and playing a computer game being successfully demonstrated. Energy harvesting from mouse sliding, patting, and bending of the device is realized with output voltages up to 37 V and power up to 48 μW while exhibiting good durability up to 20,000 cycles. This work presents a TPHNG utilizing surface charging for self-powered human behavior sensing and biomechanical energy harvesting.

## 1. Introduction

Human motion monitoring is an interesting and important field, exhibiting broad applications in medical care, search and rescue, athletic training, and antiterrorism activities [1,2,3,4]. Various devices have been exploited for human motion monitoring based on different principles, including ultrawide band, strain, piezoelectric, triboelectric, etc. [5,6,7,8]. Among these, triboelectric nanogenerators (TENG), invented in 2012 [9], have achieved great success in energy harvesting and self-powered sensing due to the features of high voltage output (up to 10 kV) and facile structure [10,11,12,13,14,15,16]. In TENGs, the functions of energy harvesting and self-powered sensing are usually closely related to each other. On the one hand, by harvesting various kinds of energies of human daily motion such as finger tapping, walking, breathing, swallowing, blinking, etc. and converting them into electricity, a sustainable power source for sensors can be fulfilled [17,18]. On the other hand, a large amount of useful data can be obtained by testing and analyzing the signals generated by human movement. Biomedical-related information has already been obtained from heartbeat, pulse, and gesture monitoring [19,20,21]. Practical device prototypes of motion monitoring TENGs have been developed, such as a TENG safety belt with a sensitivity of 0.89 V/cm^2^ for car-driving status monitoring [22,23], printing paper-based TENG with a detecting distance of 200 cm for human proximity monitoring [24], and fabric TENG with a deformation range of 90% for taekwondo motion monitoring [25].

Construction of hybrid structures consisting of various types of nanogenerators has emerged as a great trend and a boom area due to the multiple functions [26], high energy conversion efficiency [27], and remedying drawbacks of neat nanogenerators [28]. Hybrids such as triboelectric-piezoelectric [26], triboelectric-pyroelectric-piezoelectric [27], triboelectric-solar cell [28], piezoelectric-biofuel cell [29], etc., have been proposed by researchers. TENGs exhibit high sensitivity, and can be utilized for sliding-type movements sensing and harvesting [30], while their ability to sense pressing-type movements may be limited, especially for concurrently sensing the sliding and pressing motions with an individual TENG. This is because among different types of TENGs of vertical contact/separation, single-electrode, and lateral sliding, the sliding mode TENG is excellent candidate for sliding motion sensing while showing limitations for press-type motions. At the same time, piezoelectric nanogenerators (PENGs) are excellent candidates for pressing-type force sensors, with outstanding linearity and good high-frequency properties owing to the piezoelectric property of the active piezoelectric material, i.e., Q = dF, where Q is the surface charge, d is the piezoelectric coefficient, and F is the applied force [31]. Considering the common features of TENGs and PENGs, being based on surface charging of functional films, preparing TENG-PENG hybrids for concurrently sensing pressing and sliding motions represents an excellent option. Usage of computers has come to occupy a great part of daily human life; thus, the effective and precise collection of computer user behavior information is significant in various fields, such as health care, juvenile protection, safety recognition, and human–machine interfaces for artificial intelligence [32,33,34]. There are few works reporting computer user motion monitoring via nanogenerators, which is a significant topic to be explored. Thus, exploring the approach of determining a suitable device structure and materials for TENG-PENG hybrids with excellent properties of computer user sensing and energy harvesting in office environments are crucial in this field.

In this work, we investigated a hybrid nanogenerator in the form of a smart computer mousepad for the applications of computer user operation monitoring and biomechanical energy conversion. The hybrid device consists of a rubber sheet-copper trips sliding mode TENG and a poly(vinylidene fluoride-trifluoroethylene-chlorofluoroethylene) (P(VDF-TrFE-CFE)) film-based PENG. The two nanogenerators for the sliding and pressing motions sensing can work concurrently and individually, thereby boosting functions and enhancing outputs. Employing the triboelectric-piezoelectric hybrid nanogenerator (TPHNG), almost all operations involving the use of the mouse in computer usages, including single click, double click, scroll, slide, taking-up–putting-down, sliding speed, and sliding path can be recognized by their voltage signals, along with distinguishable patterns and high peak values. On the basis of operation recognitions, monitoring of computer user behavior is realized, and real-time monitoring of two different usages, namely, browsing a document and playing a computer game, are demonstrated. Furthermore, various types of human motion energies, such as sliding, hand patting, and bending can be converted into electricity by the PTHNG. The PTHNG exhibits excellent electric outputs and high energy harvesting durability. This work provides a method for preparing PTHNG for concurrent applications in behavior monitoring and energy conversion.

## 2. Materials and Methods

### 2.1. Preparation of the P(VDF-TrFE-CFE) Film and the PENG

First, 1 g of P(VDF-TrFE-CFE) (63:29:8 molar ratio, Piezotech Ltd. Pierre-Benite, France) terpolymer powder was dissolved into 10 g of dimethylformamide (99%, Aladin, Shanghai, China) and stirred for 36 h. The slurry was cast onto a flat glass with controlled area and thickness and shaped to a flat film by a doctor blade, followed by drying at 80 °C for 3 h and cured at 120 °C for 1 h in air from which solvents had been completely removed. The whole piece of P(VDF-TrFE-CFE) film with an area of 13.6 × 19.5 cm^2^ and a thickness of 20 μm was peeled off from the substrate. Two pieces of Al foils of 20 μm thickness and the same area of piezoelectric film were adhered to both surfaces of the P(VDF-TrFE-CFE) to prepare the electrodes. The P(VDF-TrFE-CFE) film was poled using a direct current (DC) power supplier at 800 V (i.e., 40 V/μm) for 24 h at room temperature. After poling, the piezoelectric responses of P(VDF-TrFE-CFE) film were examined to determine whether periodic voltages could be output upon application of compressive forces. The PENG was then sealed with polyimide films (15 × 22 cm^2^) to form a protective package.

### 2.2. Preparation of the TENG and Integration of the Hybrid Nanogenerator

A rubber sheet with a thickness of 1 mm was cut to a size of 18 × 22 cm^2^. Two rectangle copper strips 2 cm wide and 18 cm long were parallelly adhered to the center of the rubber sheet with an in-plane distance of 4 cm to form a sliding-mode TENG. The rubber sheet was covered on the top side with a layer of fiber cloth of the same size. The as-fabricated PENG was adhered to the bottom of the TENG via adhesives, with outputs of two nanogenerators connected in series to integrate it into the PTHNG as a smart mousepad. A mouse with a bottom material made of polypropylene and a bottom area of 30 cm^2^ was adopted to perform various operations such as sliding, clicking, etc., on the mousepad surface.

### 2.3. Characterization and Electric Measurement

Scanning electron microscope (SEM) measurements were performed on a JEOL JSM6480. Fourier transform infrared (FTIR) spectra were taken on an Agilent Cary 610. The d_33_ of the piezoelectric films was tested by quasi-static d_33_ m (IA ZJ-3). The open-circuit voltage (V_oc_) of the nanogenerator was measured by a digital storage oscilloscope (Tektronix TBS1072B). The short-circuit current (I_sc_) of the nanogenerator was measured by an electrometer (Keithley 6514). A force gauge (Zhisheng DS2-XD) was employed to measure the pressing force applied to the device. A linear motor (LinMot V3S4) was employed to generate reciprocating sliding, pressing, and bending movements for cyclic tests of the device.

## 3. Results and Discussion

### 3.1. Structure, Working Principle, and Electric Outputs of Hybrid Nanogenerators

We propose a PTHNG as a smart computer mousepad for computer user behavior monitoring and biomechanical energy scavenging, with the schematic of its function shown in Figure 1a. In Figure 1b, where the schematic structure of the hybrid system is shown, the system consists of a sliding mode TENG and a piezoelectric terpolymer film PENG. The sliding mode TENG was realized in a rubber sheet structure adhered with two parallel copper electrodes at the bottom with an in-plane distance of 5 cm and a mouse slider (i.e., polypropylene of its bottom) as the counterpart tribo-layer. When polypropylene slides over the copper electrodes on the rubber sheet, triboelectric current is produced between the electrodes [35]. A photograph of the bottom side of the TENG is shown in Figure 1c; the detailed physical map and an area of 18 × 22 cm^2^ can be observed, which is a typical size for a mousepad. At the bottom, a PENG with an as-poled piezoelectric terpolymer P(VDF-TrFE-CFE) film of 20 μm thick is attached to provide piezoelectric electricity output at Al electrodes upon application of pressing-type operations to the mousepad. The PENG and TENG are integrated together by adhesive and connected in series to constitute a hybrid device. Figure 1d shows a cross-sectional SEM image of the as-prepared P(VDF-TrFE-CFE) terpolymer film; dense and uniform microstructures and a film thickness of 20 μm can be observed. P(VDF-TrFE-CFE) is a terpolymer on the basis of PVDF and introduced by trifluoroethylene (TrFE) and chlorofluoroethylene (CFE) (Appendix A). It exhibits a larger dielectric constant and piezoelectric coefficient than the neat PVDF and other piezoelectric polymers, and has the potential to obtain high sensitivity [36]. Figure 1e shows a photograph of the whole sheet of P(VDF-TrFE-CFE) film. The film exhibits good transparency, flexibility, and foldability, and the complete sheet exhibits no evident defects such as fractures or pores, which is indicative of high film quality. The inset of Figure 1e shows a photograph of the sealed PENG; a well-protected package is formed, further enhancing the device reliability. Appendix A shows the FTIR spectrum of the P(VDF-TrFE-CFE) film, in which its well-polymerized feature is revealed.

The working principle of the hybrid system can be classified into two types according to various kinds of movements, i.e., sliding the mouse over the mousepad and compressing the mouse onto the mousepad. In the sliding movement, a triboelectric current i_trib_ is generated due to the potential difference between two bottom electrodes in the sliding-mode TENG. Among the complete four working states of the TENG, there are two states corresponding to the slider at the beginning and end of a slide, where no triboelectric current occurs due to being in the equilibrium states. The top part of Figure 2a shows other two states where the slider is moving through the electrodes with charges transferring and current flowing between electrodes, respectively. When the slider is moving over the left electrode on the rubber layer, the left electrode is positively charged while the right one is negatively charged due to the electrostatic induction during the friction, and the potential difference leads to i_trib_ flowing from the left electrode to the right; when the slider is moving over the right electrode on the rubber layer, the charges in the two electrode are exactly contrary to the above, generating i_trib_ flowing from the right electrode to the left. For the compressive forces, responsive electricity is mainly generated from PENG. Among the four complete states of the device, there are two equilibrium states for fully pressed and released conditions when no current flow is generated. The bottom part of Figure 2a indicates the other two states, namely, pressing and releasing. When a compressive force is applied to the device, the P(VDF-TrFE-CFE) in the PENG is compressed and the aligned dipoles are squeezed, leading to the charges on both electrodes being decreased. Thus, the piezoelectric current i_piez_ flows from the negative electrode to the positive. When the force is being withdrawn, a reverse force from the P(VDF-TrFE-CFE) film causes the film to be stretched in the vertical direction and leads to the aligned dipoles being stretched, thereby increasing the charges on both sides of electrodes, and i_piez_ flows from the positive electrode to the negative.

The electric outputs of neat TENG and neat PENG without integration were examined; Figure 2b,c shows the V_oc_ and I_sc_ of the neat TENG when sliding at a speed of 10 cm/s. It can be observed that the TENG exports steady electricity upon mouse sliding, with the output waveform taking alternative current (AC) pulsed forms with a peak V_oc_ of 46 V and a peak I_sc_ of 1.4 μA. A pair of positive and negative peaks correspond to a trip for slide movement, while two adjacent pairs of peaks correspond to two one-way sliding (i.e., forward or backward). Figure 2d,e shows the V_oc_ and I_sc_ outputs of the neat PENG on a compression force of 5 N. Steady pulsed AC electricity is generated by the PENG, with peak V_oc_ of 26 V and peak I_sc_ of 2 μA. A pair of positive and negative peaks of the outputs correspond to one pressing and releasing movement. These results indicate that the TENG and PENG are capable of independently detecting sliding and pressing movements, respectively, with good outputs.

The electric outputs of the integrated TPHNG were then investigated. Figure 2f,g shows the V_oc_ and I_sc_ outputs of the integrated mousepad with the mouse sliding over it at a speed of 10 cm/s. Stable outputs with V_oc_ of 52 V and I_sc_ of 1.8 μA are achieved. The outputs for the TPHNG are slightly higher than those for the neat TENG (V_oc_ ~ 46 V, I_sc_ ~ 1.4 μA), which indicates that there exists a coupling effect between the TENG and PENG in the hybrid system (Figure 3a,b). The phenomenon can be explained by the poled P(VDF-TrFE-CFE) in PENG producing intrinsic charges on both electrodes in PENG [37], which induces an initial charge accumulation on the copper trips in the TENG, resulting an enhancing effect similar to a charge injection [38]. On the other hand, the waveform patterns of sliding with the hybrid device are similar to those of the neat sliding mode TENG. The whole nanogenerator can generate electricity upon compressive forces, as shown in Figure 3c. For compressive forces, the hybrid device exhibits slightly higher outputs than the neat PENG. As depicted in Figure 3d, this may be due to the rubber layer in TENG playing the role of an elastic buffer layer to enhance stress transfer and leads, leading to increase piezoelectric outputs compared to the same setup without an elastic layer [39]. Generally, the hybrid nanogenerator possesses the two functions of detecting parallel slides and vertical compressive movements; and outputs are slightly enhanced in the integrated device compared to the neat TENG and PENG for both slides and compressions, indicative of a profitable coupling effect between TENG and PENG.

### 3.2. Self-Powered Mouse Sliding-Type Motion Sensing by the Hybrid Nanogenerator

In the hybrid system, the TENG is the main active part used to detect sliding movements of the mouse over the mousepad. Figure 4a shows the effect of sliding speed on the voltage output of the TPHNG. When the sliding frequency increases from 1 Hz to 11 Hz (for corresponding sliding speed of 5 to 55 cm/s, the sliding displacement is 5 cm), the voltage peak increases from 7 to 23 V due to more charge being accumulated under the higher sliding rate [40]. The speed-dependent voltage peak values indicate the application as a sliding speed sensor. Another factor reflecting the sliding speed is the pulse counts per unit time, which increases from 1 to 11 in 1 s in this sliding speed range. The comprehensive factors reflecting a parameter effectively improve the sensing accuracy [41].

In addition to sensing the sliding speed in a given direction, different sliding paths can be distinguished. Figure 4b shows the voltage output of the nanogenerator when sliding the mouse along transverse or oblique paths. At a constant speed of 10 cm/s, the voltage is higher for oblique sliding (18 V) than transverse sliding (15 V). The higher voltage is due to a larger sliding displacement, as schematically shown in Figure 4c. The result of V_oc_ increasing with the sliding displacement has been confirmed in simulated and experimental results by previous researchers [42,43]. Furthermore, direct contact between the mouse and the mousepad is not a necessary condition for the detection of mouse sliding movements. This is because the voltage signals can be exported from the device when the slider and the mousepad are in a noncontact sliding mode [43] with a small distance. As shown in Figure 5, the outputs are slightly decreased while the patterns remain unchanged.

### 3.3. Mouse Pressing-Type Motion Sensing by the Hybrid Nanogenerator

The TPHNG exhibits the ability to detect vertical pressing-type movements of the mouse by distinguishing voltage output patterns and peaks. Figure 6a shows the change in voltage output of the mousepad hybrid nanogenerator with the pressing force. For vertical pressing movements applied to the mousepad, the PENG works to export signals as a result of the P(VDF-TrFE-CFE) deforming. The peak value of voltage outputs clearly reflects the pressing force, as a higher voltage is generated at a larger compression force (10–36 V for pressing forces of 2–7 N). A strong linearity with a sensitivity of 5 V/N is obtained (inset of Figure 6a), which is due to the output nature of piezoelectric materials, i.e., Q=d·F, where, *d* is the piezoelectric constant and *F* is the applied force [44,45,46]. The d_33_ is measured to be 40.3 pC/N for the fabricated P(VDF-TrFE-CFE) film after the poling process.

For different pressing operations on the mouse, such as single clicking (right and left), double clicking, wheel scrolling, and taking-up–putting-down, the waveform pattern and peak value are different. Figure 6b,c shows the voltage output of the hybrid nanogenerator when detecting mouse left clicking and right clicking, respectively, the most common mouse operations during human usage of a computer. This set of peaks are very recognizable, with a high peak of 8 V accompanied by a low peak of 4 V for each single mouse click. The first peak corresponds to a pressing by the finger onto the mouse blade, while the latter corresponds to a rebounding of the blade. Furthermore, unlike wave patterns for mouse sliding (peak width ~ 0.033 s), the pulses for clicking are more narrow (peak width ~ 0.013 s) due to the fast speed of the blade clicking and of the reactive force involved. It can be observed that there is no evident difference in the wave pattern between a single right click and a single left click.

Figure 6d shows the voltage output corresponding to mouse double clicking, for which a clearly recognizable pattern can be found, i.e., two sets of the pattern for a single click within a brief time interval. The waveform of the voltage output of scrolling the wheel of the mouse is shown in Figure 6e. A sine wave-like voltage output with a peak value of 0.6 V is obtained for scrolling the wheel at a uniform velocity, and the voltage output pattern is distinguishable. Figure 6f shows the voltage output for taking-up–putting-down of the mouse onto the mousepad. Because this movement is similar to common pressing, the waveform pattern is similar to an ordinary single pressing operation onto the PENG, and the time interval between the positive peak and negative peak (5–10 V) is the period between taking up and putting down.

### 3.4. Computer User Bevavior Monitoring by the Hybrid Nanogenerator

For a human working at a computer, it is significant to be able to monitor human behavior by collecting data on the operations of mouse clicking, sliding, scrolling, taking-up–putting-down, etc., for process monitoring, behavior recognition, human–machine interfaces, and healthcare monitoring [32,33,34]. The phenomenon of interest is that there are various combinations of different operations engaged in by a human performing different tasks using a computer. For behavior monitoring, one premise is to precisely detect each operation, which should be confirmed by distinguished outputs for each operation in a segment of waveforms [24].

Figure 7a shows a segment of voltage output of the TPHNG containing the wave patterns for a double-click, several slides, and a single-click; all can be clearly distinguished, mainly by their feature patterns and corresponding voltage peaks. In Figure 7b, a segment of voltage output containing patterns for wheel scrolling and taking-up–putting-down; due to evident characteristics for each movement, they can be precisely recognized. Therefore, on the basis of clear distinguishment for single operations, behavior monitoring for a human working at a computer can be collected in real time.

We tested two typical usages of the mousepad, i.e., browsing a document (Appendix A) and playing a computer game (Appendix A). Figure 7c,d shows two segments of voltage output in a period of 1 min for a human performing the different tasks of browsing a document and playing a computer game, respectively. Electrocardiogram-like monitoring wave patterns have been obtained for the two tasks. Figure 7e,f shows the accumulated numbers for operations for the two tasks. For browsing a document, which usually is conducted at a moderate movement rate, there are a total of 17 clicks, 4 double-clicks, 4 taking-up–putting-downs, 20 slides and 13 scrolls in 1 min. For playing a computer game, which usually requires fast movement, 31 clicks, 10 double-clicks, 8 taking-up-putting-downs, 26 slides and 8 scrolls were performed in the same time period. These results confirm a successful detection and recognition of human behavior for working at a computer by the hybrid nanogenerator (Appendix A). The advantages of this approach to computer user behavior monitoring include its self-powered nature, facile structure, cost-effectiveness, and high precision, and it shows great potential in various applications such as user authentication and security protection, human–machine interfaces in artificial intelligence, protection of minors for avoiding overuse of computers, healthcare monitoring for general computer users, etc.

### 3.5. Biomechanical Energy Harvestingand Conversion by the Hybrid Nanogenerator

The TPHNG can be utilized for harvesting various piezoelectric and triboelectric biomechanical energy. For the piezoelectric energy harvesting, in addition to normal work at a computer, i.e., clicking and scrolling the mouse, etc., electricity can be generated upon patting/clapping the mousepad with a hand or bending the mousepad; for triboelectric energy harvesting, sliding the mouse or other dielectric objects over the pad can generate electricity via the triboelectric effect. Figure 8a–c shows the rectified V_oc_ of the hybrid nanogenerator upon sliding the mouse over the mousepad at a speed of 10 cm/s, hand patting (7 N) on the mousepad, and 180° bending of the mousepad; stable voltage output is obtained in all cases, with peaks of 37, 33, and 0.8 V, respectively. The requirement of rectified waveforms is due to the electricity transformed from AC to DC making storage easier, with the circuit structure of a full-wave bridge rectifier adopted in this study shown in Appendix A.

Figure 8d shows the load resistance dependent output power of the device for mouse sliding and hand patting. In the load resistance range of 100 kΩ–200 MΩ, the output power first increases and arrives at a peak, then decreases with the load. The maxima are obtained at the load of 1 MΩ with the largest output power of 48 and 18 μW for mouse sliding and hand patting, respectively. The peak values are obtained at 1 MΩ for both sliding and hand patting, indicative the same internal impedance of the device at the two working states due to PENG and TENG being in a series connection. Figure 8e shows charging curves of a 4.7 μF capacitor utilizing the rectified outputs of the nanogenerator upon mouse sliding and hand patting, respectively. After charging for 180 s, the capacitor exhibits a respective DC voltage of 2.87 and 1.71 V for the two movements, indicating that the nanogenerator working in conjunction with an energy storage unit is capable of driving microelectronic devices.

To demonstrate that the output electricity generated by the hybrid nanogenerator is capable of driving low power office devices, eleven commercial red LEDs were connected in series connections to the external nanogenerator through a rectification, as shown in Figure 8f. It can be found that all LEDs can be lit up by the device with the mouse sliding at a speed of 10 cm/s. As shown in Appendix A, the LEDs are turned on and off simultaneously with the sliding movements of the nanogenerator. The nanogenerator exhibits outstanding durability and reliability. We ran the TPHNG for 20,000 slidings, and its V_oc_ output is shown in Figure 8g. No noticeable change in V_oc_ output is observed, indicating outstanding reliability and stability. In the results of the compression-type cyclic test shown in Figure 8h, it can be seen that good durability is realized over 10,000 cycles of pressing with a force of 7 N. Similarly, 8000 cycles of 180° bendings (Figure 8i) were realized with stable outputs.

## 4. Conclusions

In summary, a mousepad containing a TPHNG was creatively introduced for employment in self-powered computer user behavior monitoring and biomechanical energy harvesting. The sensing and harvesting of sliding motions and compressing motions are realized by employing a sliding-mode TENG with large sensitivity and a P(VDF-TrFE-CFE) terpolymer PENG with good linearity in a hybrid nanogenerator. The outputs are slightly enhanced in the integrated device compared to the neat TENG and PENG for both sliding and compression, which is due to a profitable coupling effect between the TENG and PENG. Using the hybrid nanogenerator, almost all possible mouse operations involved in using a computer, including single clicking, double clicking, sliding, scrolling, taking up–putting down, moving speed, and moving path, were clearly tested and found to have distinguishable voltage signals with peaks from 0.6 to 36 V. Movements can be detected even with a small separation between the mousepad and the mouse, which enables precise sensing during practical use. When using the hybrid nanogenerator for behavior monitoring, data on the accumulated numbers for each operation over a 1 min period were successfully used to obtain electrocardiogram-like signals to distinguish browsing a document from playing a computer game, and the numbers for each motion were successfully obtained. In addition to being self-powering, the behavior monitoring sensor possesses the features of facile structure, cost-effectiveness, and high precision. Human motion energy harvesting is realized utilizing the hybrid nanogenerator with rectification. The energy harvesting device exhibited excellent durability for over 20,000 cycles of sliding, 8000 cycles of pressing, and 10,000 cycles of bending. Eleven LEDs could be illuminated using the power generated by the device. Our work provides a TPHNG utilizing surface charging for concurrent self-powering, human behavior sensing, and biomechanical energy harvesting in office environments.

## Figures and Tables

**Figure 1 polymers-15-02462-f001:**
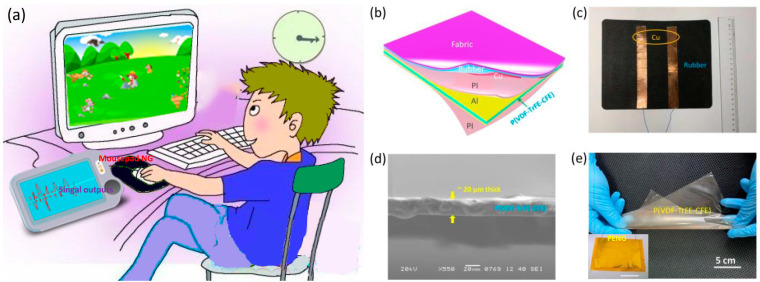
Device structure, application prospects, and images. (**a**) Schematic of smart mousepad for monitoring computer user motion on the mouse with a computer for potential applications in health care, juvenile protection, safety recognition, and human–machine interfaces for artificial intelligence. (**b**) Schematic structural illustration of the hybrid mousepad nanogenerator consists of a sliding-mode TENG and a piezoelectric terpolymer PENG. (**c**) The photograph of the back side of the as-prepared sliding-mode TENG (18 × 22 cm^2^); parallel copper strips are placed onto a rubber sheet as electrodes, and the slider, i.e., another part of the TENG, is the bottom side of the mouse. (**d**) Cross-sectional SEM image and (**e**) photograph of the as-prepared complete sheet of P(VDF-TrFE-CFE) film (13.6 × 19.5 cm^2^). The inset of (**e**) shows a photograph of an Al/P(VDF-TrFE-CFE)/Al PENG (18 × 22 cm^2^) encapsulated by polyimide layers; the scale bar is 10 cm.

**Figure 2 polymers-15-02462-f002:**
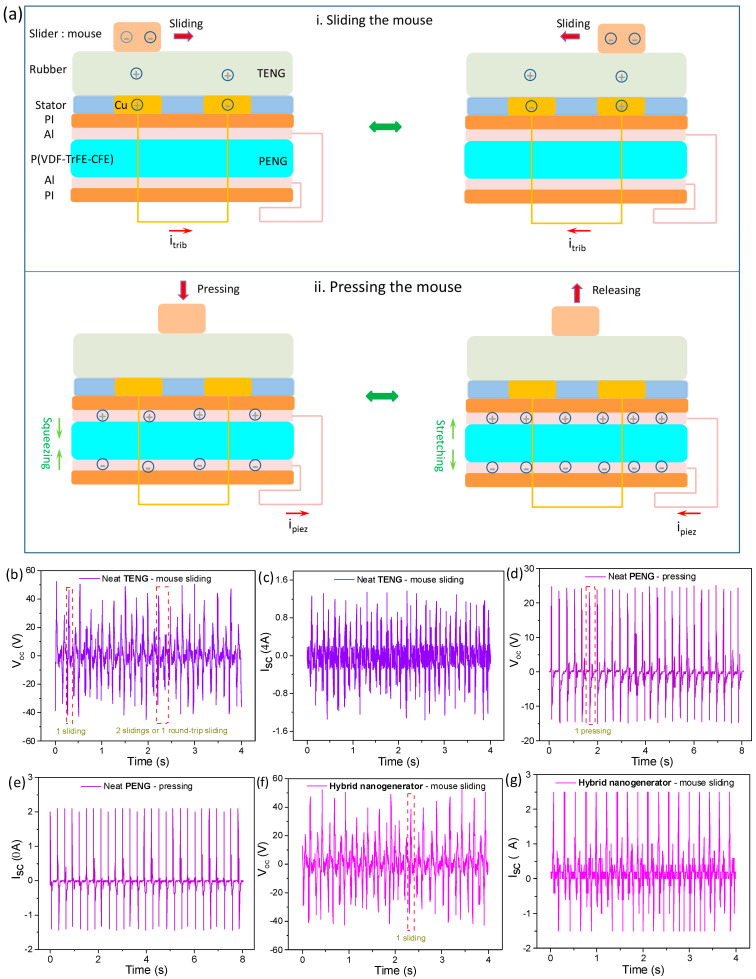
Working principle based on surface charging of functional films and electrical outputs of relative devices. (**a**) Working principles of the hybrid nanogenerator for mouse sliding-type and mouse pressing-type operations. (**b**,**c**) Voltage and current outputs of the neat sliding mode TENG under parallel sliding (~10 cm/s). (**d**,**e**) Voltage and current outputs of the neat PENG under vertical compression and releasing (~5 N). (**f**,**g**) Voltage and current outputs of the integrated device under mouse sliding over the hybrid mousepad.

**Figure 3 polymers-15-02462-f003:**
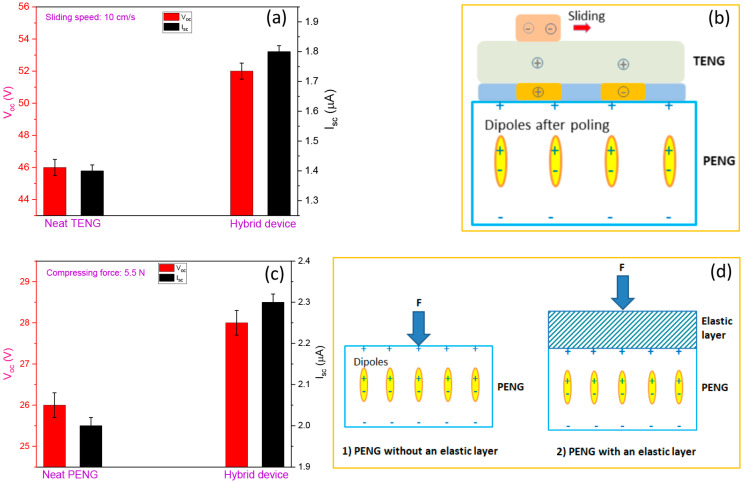
Coupling effect between PENG and TENG. (**a**) Comparison of V_oc_ and I_sc_ outputs of the neat TENG and the triboelectric-piezoelectric hybrid nanogenerator while sliding at a speed of 10 cm/s. (**b**) Schematic illustration of the effect of intrinsic charges of PENG on charge accumulation in adjacent copper electrodes in TENG. (**c**) Comparison of V_oc_ and I_sc_ outputs of the neat PENG and the triboelectric-piezoelectric hybrid nanogenerator under 5.5 N compression. (**d**) Schematic illustration of the effect of the elastic layer on stress transfer in the PENG.

**Figure 4 polymers-15-02462-f004:**
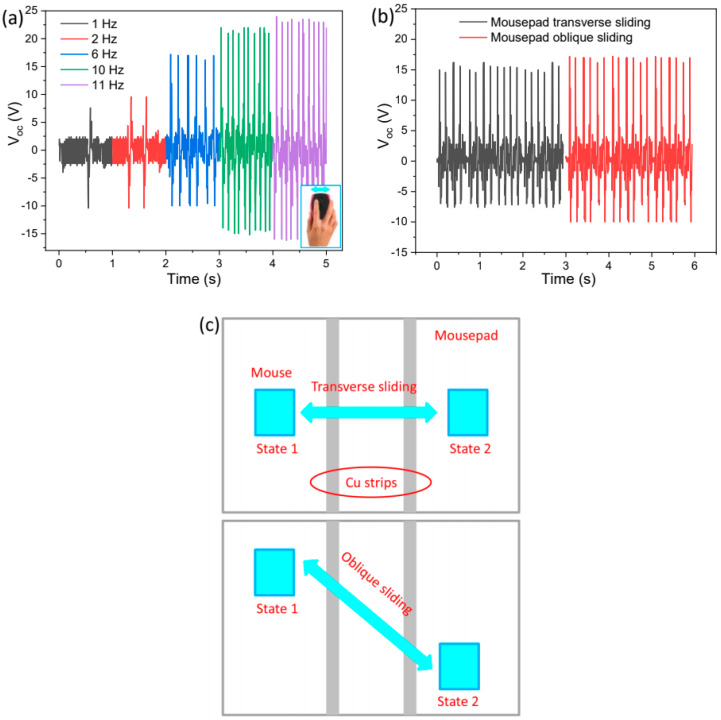
Self-powered mouse lateral sliding operation sensing. (**a**) Voltage outputs of the hybrid mousepad for sliding the mouse over it with various speeds. The inset of (**a**) shows an image of a mouse sliding operation. (**b**) Voltage outputs of the hybrid mousepad using different sliding paths. (**c**) Schematic of the various sliding paths of the mouse over the hybrid nanogenerator mousepad.

**Figure 5 polymers-15-02462-f005:**
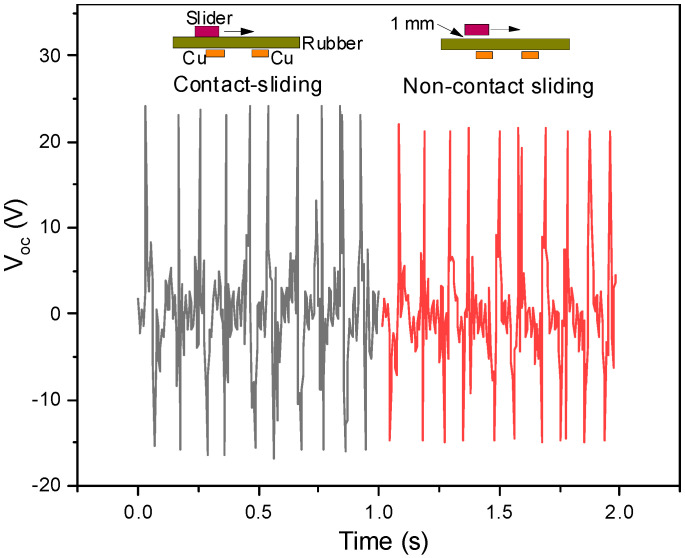
Voltage outputs of the hybrid nanogenerator for contact-sliding and non-contact sliding with a small separation between the slider and the rubber sheet.

**Figure 6 polymers-15-02462-f006:**
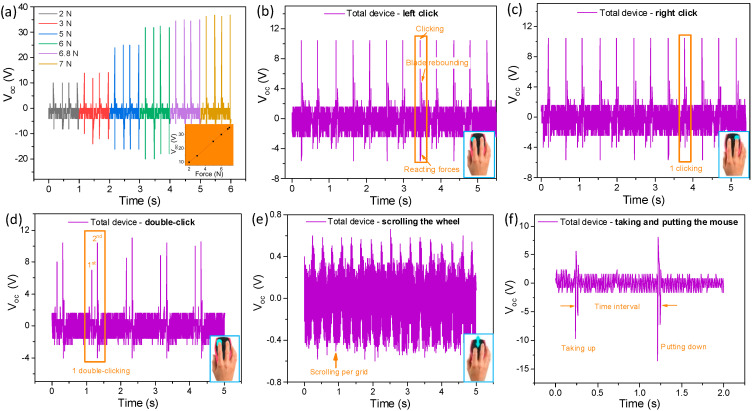
Self-powered mouse clicking operation sensing. (**a**) Voltage outputs of the smart mousepad as a function of compressive forces. The inset of (**a**) shows the plot of the force–voltage relationship. (**b**–**f**) Voltage output patterns for mouse left clicking, right clicking, double clicking, wheel scrolling, and hitting (i.e., taking-up–putting-down). The insets in (**b**–**e**) show schematic plots for the various operations.

**Figure 7 polymers-15-02462-f007:**
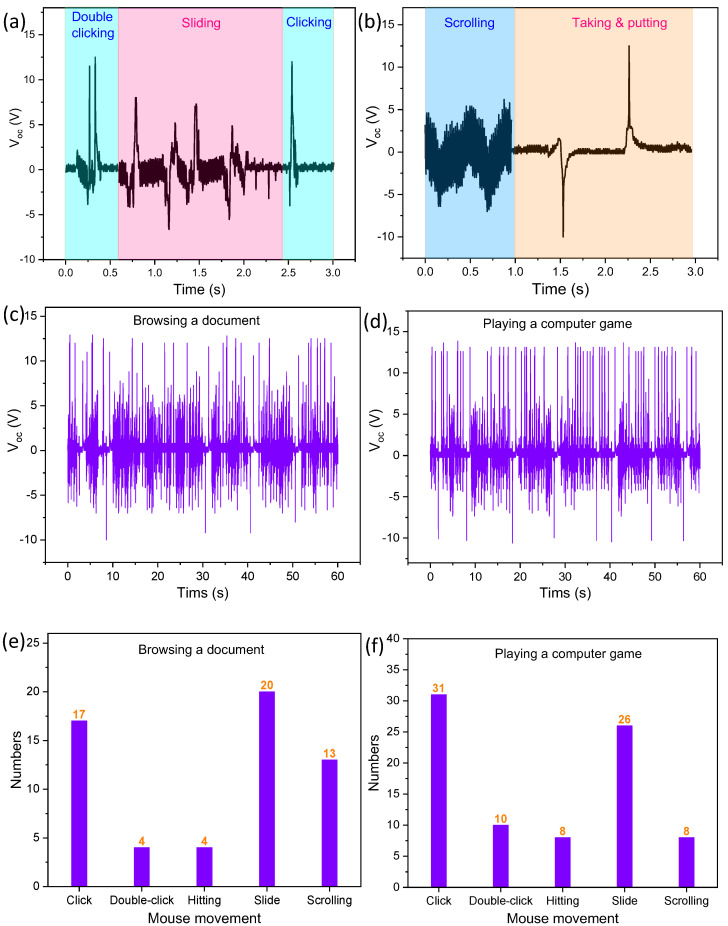
Self-powered computer user behavior monitoring. (**a**,**b**) Various mouse operations distinguished from voltage signals generated by the mousepad under taking-up–putting-down, single clicking, double clicking, scrolling, and sliding. (**c**,**d**) A 1 min segment of the voltage signal pattern of the hybrid nanogenerator for the computer user in (**c**) browsing a document and (**d**) playing a computer game. (**e**,**f**) The counting of different operations of computer user behavior in (**e**) browsing a document and (**f**) playing a computer game for a period of 1 min.

**Figure 8 polymers-15-02462-f008:**
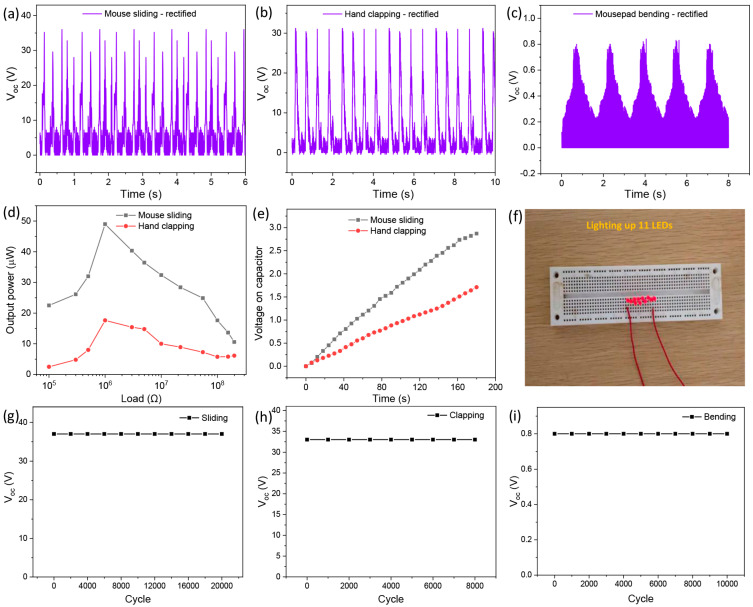
Biomechanical energy harvesting. (**a**–**c**) Full-wave bridge rectified voltage outputs of the hybrid nanogenerator by (**a**) mouse sliding, (**b**) hand patting, and (**c**) 180° pad bending. (**d**) Changes in output power of the hybrid nanogenerator with load resistance for mouse sliding and hand patting, respectively. (**e**) Charging voltage curves of a 4.7 μF capacitor by the rectified output of the nanogenerator for mouse sliding and hand patting, respectively. (**f**) Photograph illustrating the lighting up of eleven LEDs by the mousepad nanogenerator for mouse sliding. (**g**–**i**) V_oc_ durability of the mousepad nanogenerator with 20,000 slidings, 8000 pressings, and 10,000 bendings.

## Data Availability

Data will be made available on request.

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
