# Peer review of "A Mousepad Triboelectric-Piezoelectric Hybrid Nanogenerator (TPHNG) for Self-Powered Computer User Behavior Monitoring Sensors and Biomechanical Energy Harvesting"

_polymers, 2023, doi:10.3390/polym15112462_

Round 1

Reviewer 1 Report

The authors studied the triboelectric-piezoelectric  hybrid nanogenerator for computer user to achieve energy harvesting. In general, this is an interesting paper. However, there are some questions/issues needed to be addressed before being published:

  1. I would recommend the authors include some more technic background details within the introduction section such as previous studies for TENG, PENG and TPHNG. Currently, it mainly cover the non-technic part.
  2. Between line 91-92, the authors mentioned the film was left to be polymerized. However, I do not think it is accurate, It was polymer before casting and there should be no polymerization involved within the casting process.
  3. In addition, the casted film was left under high temperature for more than 1h. Is the purpose of this step to fully remove the solvent or to cast a uniform film?
  4. Following on last point, it has been reported previously that the DMF can induce the degradation of PVDF, particularly at a high temperature. Given the image of the cast film, I would assume there are some degradation happening during the casting process. Please comment on it.
  5. Please provide some more details regarding the poling process. In addition, I wonder how the authors determine the completion of the poling?
  6. PENG and TENG were laminated as mentioned by the authors. I wonder if the authors have considered the energy loss by the lamination layer? If yes, could the authors estimated the percentage of loss as well as the solution to improve it?
  7. For Figure 1a, did the authors make it or have the permission using it? If not, I may have some concerns regarding the copy right.
  8. For Figure 2a, the authors claimed PVDF for the PENG. However, I believe it should be the terpolymer. In addition, the charges seem misleading given it should only be the dipole moment alignment instead charges.
  9. For Figure 7, the plots seems not very informative to me. In addition, the authors did not demonstrate its representativity. I would recommend either the authors move it into the supporting information and/or try to obtain some more representative data.
  10. The plots within Figure 8 g - I are missing.
  11. I noticed different compression forces were used within the study (5N in Figure 2 and 7N for Figure 8 and etc.) I wonder why the authors not used consistent forces within the study?

Author Response

Referee 1:

The authors studied the triboelectric-piezoelectric  hybrid nanogenerator for computer user to achieve energy harvesting. In general, this is an interesting paper. However, there are some questions/issues needed to be addressed before being published:

  1. I would recommend the authors include some more technic background details within the introduction section such as previous studies for TENG, PENG and TPHNG. Currently, it mainly cover the non-technic part.

Reply: Thanks for the suggestion. More technic backgroud details on TENG, PENG and TPHNG are provided in the revision.

  1. Between line 91-92, the authors mentioned the film was left to be polymerized. However, I do not think it is accurate, It was polymer before casting and there should be no polymerization involved within the casting process.

Reply: Thanks for pointing out this point. This sentence is modified: “The slurry was casted onto a flat glass with controlled area and thickness and shaped to a flat film by a doctor blade, followed by dried at 80 °C for 3 h and cured at 120 °C for 1 h in air to get polymerized.”

  1. In addition, the casted film was left under high temperature for more than 1h. Is the purpose of this step to fully remove the solvent or to cast a uniform film?

Reply: The purpose of this step is to fully remove the solvent.

  1. Following on last point, it has been reported previously that the DMF can induce the degradation of PVDF, particularly at a high temperature. Given the image of the cast film, I would assume there are some degradation happening during the casting process. Please comment on it.

Reply: Compared to other solvents, the selection of DMF is mainly due to that it is able to form a very clear solution of PVDF, and also, the DMF solution possesses good wettability with the glass plate to form a uniform PVDF film. To the best of our experience, the damage from DMF to the PVDF in our study is limited and small.

  1. Please provide some more details regarding the poling process. In addition, I wonder how the authors determine the completion of the poling?

Reply: The P(VDF-TrFE-CFE) film was directly poled using a direct current (DC) power supplier at 800 V (i.e., 40 V/μm) for 24 h at room temperature. After the poling, the piezoelectric responses of P(VDF-TrFE-CFE) film were examined, if perodic voltages can be outputed upon compressive forces. The voltage is elaborately selected to fully polarize the P(VDF-TrFE-CFE) film and also to avoid breakdown by excessive voltages. The poling process is also based on the experience and knowledges of our previous similar studies (Ref. 31).

  1. PENG and TENG were laminated as mentioned by the authors. I wonder if the authors have considered the energy loss by the lamination layer? If yes, could the authors estimated the percentage of loss as well as the solution to improve it?

Reply: Thanks for the comment. The energy loss in the lamination layer is very small. On the other hand, there is a profitable coupling between the layered TENG and PENG, which leads to an output enhancement for both TENG and PENG (Fig. 3).

  1. For Figure 1a, did the authors make it or have the permission using it? If not, I may have some concerns regarding the copy right.

Reply: Thanks for kind reminding of the reviewer, the plot of Figure 1a was drawn by the authors, so there is no copy right issue.

  1. For Figure 2a, the authors claimed PVDF for the PENG. However, I believe it should be the terpolymer. In addition, the charges seem misleading given it should only be the dipole moment alignment instead charges.

Reply: Thanks for pointing out this issue. PVDF should be P(VDF-TrFE-CFE), which has been modified in the updated Figure 2a. Concerning of the piezoelectric response, the authors would like to explain that the inner mechanism is the changing of length of alignment dipole moment, and the surface charges are the resultant phenomenon. Corresponding discussions are provided in the manuscript: “When a compressive force is applied to the device, P(VDF-TrFE-CFE) in the PENG is compressed and aligning dipoles are being squeezed, leading to charges on both electrodes to decrease.”

  1. For Figure 7, the plots seems not very informative to me. In addition, the authors did not demonstrate its representativity. I would recommend either the authors move it into the supporting information and/or try to obtain some more representative data.

Reply: Thanks much for the suggestion. The authors think Figure 7 is very important for illustrating the function of the device. The authors would like to explain that we used the similar drawing fashion to that in Ref. 24.

  1. The plots within Figure 8 g - I are missing.

Reply: Thanks for pointing out this confusing part. In fact, this is due to the too dense of the output curves to be clearly observed. New figures for Figure 8 g - I are provided.

  1. I noticed different compression forces were used within the study (5N in Figure 2 and 7N for Figure 8 and etc.) I wonder why the authors not used consistent forces within the study?

Reply: In energy harvesting situation (Figure 8), a harder compressive force (5N) may be used to generate a higher electric output. In Figure 6a, compressive forces from 2 N to 7 N can be normally used for the PENG unit in the device.

Reviewer 2 Report

Please find the review file.

Author Response

Response to reviewer comments

According to comments of the reviewer, revisons were carefully made. Comments are replied point by point as below.

Referee comment

Referee 2:

This work proposes a polymer based mousepad triboelectric-piezoelectric hybrid nanogenerator (TPHNG) for user behaviour monitoring and biomechanical energy. The topic is certainly of interest even if the application of the sensor is not innovative, the results achieved are presented good but concerning. The authors proposed and assessed an experimental prototype. The manuscript requires major changes in the English, the obtained results need to be improved and additional comparison table needs to be added to organize it satisfactory. I have following concerns for the author. Please follow below.

  1. While I appreciate the idea of behaviour monitoring, I do have some serious concerns regarding the use of sensors for biomechanical energy. The claim made about biomechanical energy is worrisome, and I suggest that the author focuses on one specific application to address this issue.

Reply: Thanks much for the appreciation on the behaviour monitoring part of our manuscript. The authors would like to explain that for both TENG and PENG, they do have both sensing and energy harvesting functions. So for the hybrid nanogenerator, it also possesses the same function simultaneously. That is the original thought of this work. In fact, the sliding motion, the most frequent motion of the mouse will generate remarkable energy from the TENG unit of the device; if a harder force is used to input compressive force to the PENG unit, remarkable energy can also be generated. And the authors do think both behaviour monitoring and energy harvesting are resonable.

  1. I am not convinced about the motivation for the presented prototype. Could you please explain how it differs from existing technologies?

Reply: It is a hybrid sensor and hybrid energy harvester. As a hybrid sensor, it is able to efficiently detecting sliding-type and compressive-type movements of mouse motions simultaneously. As a hybrid energy harvester, it possesses higher energy harvesting efficiency due to the multi-functions and high efficiencies. This device is novel, no such stucture has been reported yet, to the best of our knowledge.

  1. Although there have been similar works and prototypes presented, such as the biomechanical energy from the foot and pedestrian floor, this technology does not generate a consistent power output.

Reply: The energy output is related to the input. As for the energy harvester in this study, the input is mainly conducted by hand, so the output is relatively smaller than biomechanical energy from the foot in literatures.

  1. I have some doubts about the claim made in the paper that a single person gaming can generate the amount of power mentioned. It would be helpful to see a small video to verify this claim. This video need not be included in the paper, but it would be appreciated for the review process.

Reply: A video (Supporting Video S2) is provided in the supplementary file, from which 11 LEDs driven by the output electricity of the nanogenerator upon sliding movements can be seen.

  1. Can you please explain how this technology is superior to the copper tap or other piezoelectric sensors?

Reply: As for mouse movements, there are very many movements, sliding type and compressive type. This sensor can fulfill the complete sensing of mouse movements, while for copper tap or piezoelectric sensors, they can not achieve this purpose.

  1. My last concern is the references needs be added further which are relevant to sensors. In conclusion I think it would be better if author rethink about the application. There are several applications you can suggest it and include the comparison table. Now this work is above for the editor.

Reply: According to reviewer’s comments, more references on sensors (Ref. 45, Ref. 46) are added in the revision.

Round 2

Reviewer 1 Report

Thanks for the updated manuscript. I still have some questions regarding the updated version:

1. I do not the added part support enought technical background of this study. Only a few data do not really address my concerns. I would suggest the authors to discuss the differences and principals of TENG and PENG.

2. The descriptions for the casting seem inaccurate still. For instance, between line 82 and 83, polymerization was used. Would the authors help me to understand how the P(VDF-TrFE-CFE) (a polymer) get polymerized? Or were authors trying to say crosslinked? If so, what is the crosslinking agent?

Some minor formatting issues

Author Response

Response to reviewer comments

Referee #1 comment:

Thanks for the updated manuscript. I still have some questions regarding the updated version:

  1. I do not the added part support enough technical background of this study. Only a few data do not really address my concerns. I would suggest the authors to discuss the differences and principals of TENG and PENG.

Reply: Thanks for the comment. Discussions on the differences and principals of TENG and PENG were added in the introduction section.

  1. The descriptions for the casting seem inaccurate still. For instance, between line 82 and 83, polymerization was used. Would the authors help me to understand how the P(VDF-TrFE-CFE) (a polymer) get polymerized? Or were authors trying to say crosslinked? If so, what is the crosslinking agent?

Reply: It is a curing process with solvent gradually gone and solid phase staying. The sentence was changed: “dried at 80 °C for 3 h and cured at 120 °C for 1 h in air in which solvents were completely removed”.

Reviewer 2 Report

There are some minor English grammar mistakes to be taken care of. 

There are some minor English grammar mistakes to be taken care of. 

Author Response

English grammar mistakes were corrected